**Data Availability Statement:** All relevant data are within the manuscript and its Supporting information files.

# Elevated neutrophil-to-lymphocyte ratio and predominance of intrahepatic cholangiocarcinoma prediction of poor hepatectomy outcomes in patients with combined hepatocellular–cholangiocarcinoma

Tai-Jan Chiu[1,2,3], Yi-Ju Chen[4], Fang-Ying Kuo[5], Yen-Yang Chen[1,2]*

1 Division of Hematology-Oncology, Department of Internal Medicine, Kaohsiung Chang Gung Memorial Hospital, Kaohsiung, Taiwan and Chang Gung University College of Medicine, Kaohsiung, Taiwan, 2 Kaohsiung Chang Gung Cholangiocarcinoma and Pancreatic Cancer Group, Cancer Center, Kaohsiung Chang Gung Memorial Hospital, Kaohsiung, Taiwan, 3 Institute of Clinical Medical Sciences, Chang Gung University, Kaohsiung, Taiwan, 4 Department of Anatomic Pathology, E-Da Hospital, I-Shou University, Kaohsiung, Taiwan, 5 Department of Pathology, Kaohsiung Chang Gung Memorial Hospital, Chang Gung University College of Medicine, Kaohsiung, Taiwan

* chen.y9964@gmail.com

## Abstract

### Objectives

Although elevated neutrophil-to-lymphocyte ratio (NLR) has been associated with survival in some liver cancers, its prognostic relevance has not been studied in the context of combined hepatocellular cholangiocarcinoma CHCC-CC, a rare primary liver cancer. We investigated whether elevated NLR and a predominance of cholangiocarcinoma might predict poor prognosis in patients with resectable CHCC-CC.

### Methods

We retrospectively reviewed the clinicopathologic data of forty-two patients with CHCC-CC receiving hepatectomies at our hospital. We used Kaplan-Meier and Cox regression to analyze survival.

### Results

Two-year disease-free survival and five-year overall survival rates were 43.2% and 32.9%, respectively. Univariate analyses showed that patients with NLR $\geq$3 had significantly worse 2-year DFS and 5-year OS rates. Univariant Kaplan-Meier survival analysis also associated these rates with a predominance in intrahepatic cholangiocarcinoma, AJCC tumor stage, pathological T stage and lymph-vascular invasion. However, our multivariate analysis found NLR $\geq$3 to be the only independent predictor of disease recurrence and poorer survival.

**Funding:** This study was funded by grant obtained by Dr. Yen-Yang Chen from Kaohsiung Chang Gung Memorial Hospital Taiwan (Grant nos. CMRPG8E0881, CMRPG8F1771/1772). This study was also funded by grants obtained by Dr. Tai-Jan Chiu from Kaohsiung Chang Gung Memorial Hospital Taiwan (Grant CMRPG8E0811/0812/0813, CMRPG8C0531 and CMRPG8D0801).

**Competing interests:** The authors have declared that no competing interests exist.

**Abbreviations:** AFP, Alpha-Fetoprotein; CEA, Carcinoembryonic antigen; CHCC-CC, combined hepatocellular cholangiocarcinoma; CK7, cytokeratin 7; CT, computed tomography; DFS, disease free survival; ECOG, Eastern Cooperative Oncology Group; H&E, Hematoxylin and Eosin; HbsAg, surface antigen of the hepatitis B virus; HCC, Hepatocellular carcinoma; HCV, hepatitis C virus; Hep1, hepatocyte paraffin 1; HR, hazard ratios; ICG, indocyanine green; IHC, intrahepatic cholangiocarcinoma; MR, magnetic resonance; NLR, neutrophil-to-lymphocyte ratio; OS, overall survival; PS, performance status.

## Conclusions

Neutrophil-to-lymphocyte ratio was the most important independent predictor of poorer survival in patients with resectable CHCC-CC. Predominance of intrahepatic cholangiocarcinoma, advanced AJCC tumor stage and pathological T stage, and lymph-vascular invasion also may affect poor prognosis in patients receiving complete tumor resections.

## Introduction

Combined hepatocellular cholangiocarcinoma (CHCC-CC) is an uncommon primary liver malignancy. It is unique in that it shares pathological characteristics of hepatocellular carcinoma (HCC) and intrahepatic cholangiocarcinoma (IHC) coexisting within the same tumor. It makes up between 1% to 14.3% of all primary liver tumors depending on the study [1–9]. It is difficult to use image studies to accurately diagnose this disease prior to non-invasive preoperative examinations [10]. Currently, complete tumor resection is the only possible curative treatment.

Although some studies have attempted to identify factors that can predict post-hepatectomy disease free survival and overall survival, their results have not been in agreement. While some of them have reported that CHCC-CC has a worse median survival than hepatocellular carcinoma (HCC) and intrahepatic cholangiocarcinoma (IHC) [3, 9, 11], others report its survival rate to fall between the survival rates of HCC and IHC [12–14] or be equal to IHC but worse than HCC [15]. One study found IHC-dominant patients with recurrent tumors to have significantly shorter mean survival rates than those with HCC-dominance [16]. Thus, it remains unclear how a predominance of HCC cells vs. IHC cells might affect surgical outcomes in patients with CHCC-CC.

Tumors induced systemic inflammatory cytokines and mediators could promote angiogenesis, DNA damage, and inhibition of apoptosis of cancer cells and associated with poor prognoses in different cancers [17–19]. While tumor was diagnosed and before treatments, inflammatory response could be easily detected by peripheral blood tests. Several peripheral blood parameters, including neutrophil-to-lymphocyte ratio (NLR), derived NLR (d-NLR), platelet-to-lymphocyte ratio, and monocyte-to-lymphocyte ratio, have been found to be indicators of systemic inflammation and have been found predict prognosis in several malignancies [20, 21]. Although some studies have associated elevated NLR with poorer prognosis in patients with intrahepatic cholangiocarcinoma and hepatocellular carcinoma [22, 23], very few have discussed have its relevance to prognosis of CHCC-CC patients after hepatectomy.

Therefore, this study was conducted to evaluate the possible association of systemic inflammation (NLR) and other pathological factors with disease free and overall survival in CHCC-CC patients after tumor resection.

## Materials and methods

### Patients

This study was approved by the Chang Gung Medical Foundation Institutional Review Board, the case number is 103-7412B. All methods were performed in accordance with the principles expressed in the Declaration of Helsinki. Written informed consent was obtained from all patients before surgery. All data were fully anonymized before we accessed them. Patients' medical records were accessed between January 2000 and December 2013.

We retrospectively reviewed the records of forty-two patients diagnosed as having CHCC-CC post hepatectomy between January 2000 and December 2013 at Kaohsiung Chang Gung Memorial Hospital in Kaohsiung a medical center located in southern Taiwan. All of these patients had intrahepatic CHCC-CC post complete tumor resections (M0) and no synchronous primary tumors. CHCC-CC was diagnosed pathologically based on microscope studies of thin-section specimens stained with hematoxylin and eosin. The immunoreactivity of each tumor was confirmed [24]. They only had type B combined disease (contiguous but independent masses of HCC and IHC) and type C combined disease (intimate intermingling of hepatocellular and glandular elements), based on Allen and Lisa classification. None of the patients had type A (separate masses constituting either HCC or IHC). None of the patients had thrombus extending to the level of the superior mesenteric vein or any antitumor treatments before operation. In addition, all patients had to be ≥18 years old and have a performance status (PS) of ≤2 on the Eastern Cooperative Oncology Group (ECOG) scale, adequate bone marrow, hepatic and renal function (creatinine clearance >60 mL/min), and a computed tomography or magnetic resonance image scan of the abdominal region within three weeks prior to the initiation of treatment.

Before hepatectomy, each patient received ultrasonography, contrast-enhanced magnetic resonance (MR), and/or tri-phase contrast enhanced helical computed tomography (CT) for the diagnosis of CHCC-CC and evaluation of the tumor. Also prior to operation, the liver function of each patient was assessed using the Child-Pugh classification system and/or the indocyanine green (ICG) clearance test. Three days before hepataectomy, neutrophil and lymphocyte counts were measured. NLR was calculated by dividing the neutrophil measurement by the lymphocyte measurement.

Clinicopathological information including age, gender, tumor (T) stage, nodal (N) status, TNM stage, and survival was obtained from the patients' clinical records. This study was approved by the Chang Gung Medical Foundation Institutional Review Board. No written informed consent was required due to the retrospective design of the study.

## Pathologic examination

As our literature review, there was no study to evaluate whether the predominance of intrahepatic cholaginocarcinoma affected the outcome in resectable CHCC-CC patients. After discussion with our pathologist with expertise in hepatic tumors, we made the definition of predominance of intrahepatic cholangiocarcinoma while those whose IHC cells made up more than 55%. Patients with CHCC-CC were subdivided into those whose IHC cells made up 55% or less (n = 23) of all cells within the largest tumor and those whose IHC cells made up more than 55% (n = 19). Two pathologists reviewed all pathological findings and concluded that both HCC cells and IHC cells coexisted in liver tumors of our patients and that IHC cells made up at least more than 10% of the cells within the tumor in all patients. Pathological diagnosis of CHCC-CC was made microscopically using thin-section specimens stained with Hematoxylin and Eosin (H&E). Immunohistochemical staining of each tumor was also performed. HCC cells were confirmed based on immunohistological stains of hepatocyte paraffin 1 (Hep1) and ICC cells confirmed based on cytokeratin 7 (CK7).

## Definition of NLR

NLR was calculated by dividing the absolute neutrophil count by the absolute lymphocyte count. NLR was determined within seven days before hepatectomy. Based on the time-dependent receiver operating characteristic (ROC) curve of our study, we set the NLR cutoff value to

be 3.0(NLR < 3, NLR ≥ 3) (S1 Fig). Previous studies in intrahepatic cholangiocarcinoma have used a similar threshold [25, 26].

### Follow-up

Patients were followed up every two months the first three years post surgery and every six months thereafter. Disease free survival (DFS) was defined as the duration between hepatectomy and recurrence or between hepatectomy and the last observation for patients without recurrence. Overall survival was defined as the time between hepatectomy and death or between hepatectomy and last follow-up.

### Statistical analyses

The differences in numerical data between the two groups were compared by Student t test. Categorical variables were compared using χ2-tests. The estimated OS was calculated using Kaplan-Meier, and the differences between groups were assessed using the log rank test. Univariate and multivariate analyses of the prognostic factors for survival were performed using the Cox proportional hazards model. All statistical operations were performed using SPSS 18 statistical software (IBM Corp., Armonk, NY, USA). All tests were two-sided. A p-value < 0.05 was considered significant.

## Result

### Clinicopathological characteristics

The median age of 42 patients whose cases we reviewed was 58 years (range, 32–80 years). Twenty-nine (69%) were male and 13 (31%) female. The demographic and case characteristics of the patients are shown in Table 1. Classified using the TMN cancer staging system (American Joint Committee on Cancer), 15 patients (35.7%) had stage I tumors, 19 (42.5%) stage II, 4 (9.5%) stage III and 4 (9.5%) stage IVA. No patient had stage IVB (distant metastasis (M1)) disease. Only three patients had regional lymph nodes metastases (N1) and only one patient had tumor with portal vein invasion (T4). Thirty patients (71.4%) tested positive for HbsAg (surface antigen of the hepatitis B virus), 12 (28.6%) positive for HCV antibody (hepatitis C virus), and 31 (72.8%) had liver cirrhosis. Preoperatively, 23 patients (54.8%) had AFP (Alpha-Fetoprotein) levels > 15 ng/ml, three patients (7.1%) had CEA (carcinoembryonic antigen) > 5 ng/ml, and twelve patients (28.6%) CA 19–9 > 37 U/ml in. Nineteen patients (45.2%) had a predominance of IHC cells within the tumor.

### Correlation between clinicopathologic parameters and NLR

Patients were subdivided into those with NLR<3 (n = 21, 50%) and those with NLR≧3 (21, 50%). The two NLR groups had no significant differences with regard to age, gender, AFP, CA19-9, HbsAg, HCV Ab, liver cirrhosis, lymph node metastasis, perineural invasion, lymph-vascular invasion, histology grade, AJCC tumor stage or IHC cell predominance (Table 2).

### Surgical outcomes

Mean survival follow-up was 677 days (87–2371 days). Nineteen of the patients died during the course of this study. Mean disease free survival was 499 days, two-year recurrence free survival was 43.2%, and five-year overall survival 34.9% (Table 3). As can be seen in Fig 1, the following were found by univariate analyses to be associated with poor disease free survival: lymph-vascular invasion (P = 0.035, Fig 1A), IHC cell > 55% (P = 0.031, Fig 1B), AJCC stage III/IV (P < 0.001, Fig 1C), T3/4 disease (P < 0.001, Fig 1D), NLR ≧3 (P<0.001, Fig 1E).

**Table 1.  Clinicopathologic features of 42 patients with resectable combined hepatocellular cholangiocarcinoma post operation.**

| Parameters | No. of cases (percentage) |
| --- | --- |
| Age (years)(mean: 58.2, median: 58.5, range 32–80) | |
| Age≦60 | 22 (52.3%) |
| Age > 60 | 20 (47.6%) |
| Sex | |
| Male | 29 (69%) |
| Female | 13 (31%) |
| Clinical 7th AJCC stage | |
| I-II | 34 (81%) |
| III-IV | 8 (19%) |
| pathological T classification | |
| T1 | 15 (35.7%) |
| T2 | 21 (50%) |
| T3 | 5 (11.9%) |
| T4 | 1 (2.4%) |
| Pathological N classification | |
| negative | 39 (92.9%) |
| positive | 3 (7.1%) |
| Surgical margin | |
| <10 mm | 18 (42.8%) |
| ≥ 10 mm | 24 (57.2%) |
| Lymph-vascular invasion | |
| negative | 26 (61.9%) |
| positive | 16 (38.1%) |
| Perineural invasion | |
| negative | 39 (92.9%) |
| positive | 3 (7.1%) |
| Histology grade | |
| Well or Moderate | 35(83.3%) |
| Poorly | 7 (16.7%) |
| Hepatitis B | |
| Negative | 12(28.6%) |
| positive | 30(71.4%) |
| Hepatitis C | |
| Negative | 30(71.4%) |
| positive | 12(28.6%) |
| Liver cirrhosis | |
| no | 11(26.2%) |
| yes | 31(72.8%) |
| CEA | |
| ≤5 | 39(92.9%) |
| >5 | 3(7.1%) |
| CA-199 | |
| ≤ 35 | 30(71.4%) |
| >35 | 12(28.6%) |
| AFP | |
| ≤ 15 | 19(45.2%) |

(*Continued*)

**Table 1.** (Continued)

| Parameters | No. of cases (percentage) |
|---|---|
| >15 | 23(54.8%) |
| NLR | |
| <3 | 21(50%) |
| ≧3 | 21(50%) |
| Intrahepatic cholangiocarcinoma cell | |
| ≤55% | 23 (54.8%) |
| >55% | 19 (45.2%) |

Abbreviations: AFP, Alpha-Fetoprotein; CA19-9, cancer antigen 19–9; LMR, lymphocyte to monocyte ratio; NLR, neutrophil-to-lymphocyte ratio.

Overall survival was also significantly lower in patients with lymph-vascular invasion (p = 0.001, Fig 2A), IHC cell > 55% (P = 0.034, Fig 2B), AJCC tumor staging III/V (p < 0.001, Fig 2C), T3/4 disease (p < 0.001, Fig 2D), NLR ≧3 (p < 0.001, Fig 2E).

### Prognostic significance of NLR

The clinicopathological characteristics were included in our univariate and multivariate analyses. As shown in Table 3, in our univariate analysis, lymph-vascular invasion, IHC cell percentage, AJCC tumor stage, T stage and NLR were all significantly associated with DFS and OS. Multivariate cox proportional hazard regression analysis, however, revealed that only NLR ≧3 to independently predict poor DFS (HR, 41.679; 95% CI, 7.812–222.378; P = 0.001) and OS (HR, 4.148; 95% CI, 1.196–14.388; P = 0.025) in our CHCC-CC after hepatectomy (Table 4).

## Discussion

Combined hepatocellular cholangiocarcinoma is a rare mixed primary liver malignancy in which HCC and IHC coexist. It is important to identify prognostic factors at the time of diagnosis because this information may help operative decision making and guide adjuvant or neoadjuvant treatment choices. This study found that an elevated NLR (≧3.0) independently predicted worse DFS and OS in patients with CHCC-CC treated with curative resection. NLR can be calculated from a simple blood test and can be assessed prior to the initiation of therapy.

Previous studies have demonstrated an association between NLR and survival of IHC [22, 26], HCC [27, 28] and CHCC-CC [29]. Although the prognostic significance of inflammation-based NLR has been previously reported in various types of cancer, it remained unclear whether these scores might predict which patients might be at high risk of recurrence after surgery for CHCC-CC. Our current study found that NLR can also be used to predict this recurrence and overall survival.

In the microenvironment of tumors, the mediators and cellular effectors of inflammation are important constituents [30]. Inflammation may promote the development, progression, angiogenesis, invasion and metastasis of some types of cancer [31, 32]. Inflammation activates proangiogenic factors including vascular endothelial growth factor or inflammatory cytokines, such as IL-1β by increasing neutrophils [33, 34]. In the inflammatory process, lymphocytes play a very important role in innate immunity and adaptive immune response and can eradicate tumor cells by inhibiting cell proliferation or migration [17, 30]. Thus, NLR is a very practical biomarker marker of inflammation.

**Table 2.** Relationships between elevated neutrophil-to-lymphocyte ratio and clinicopathological factors.

| NLR | <3 | ≧3 | P |
|---|---|---|---|
| **Age** | | | |
| ≤ 60 | 11 (50%) | 11(50%) | 1.000 |
| > 60 | 10 (50%) | 10 (50%) | |
| **Gender** | | | |
| male | 14 (48.3%) | 15 (51.7%) | 1.000 |
| female | 7 (53.8%) | 6 (46.2%) | |
| **AJCC staging** | | | |
| I-II | 19(55.9%) | 15 (44.1%) | 0.238 |
| III-IV | 2 (25%) | 6 (75%) | |
| **T stage** | | | |
| I-II | 20 (55.6%) | 16 (44.4%) | 0.184 |
| III-IV | 1 (16.7%) | 5 (83.3%) | |
| **N stage** | | | |
| Negative | 20 (51.3%) | 19 (48.7%) | 1.000 |
| positive | 1 (33.3%) | 2 (66.7%) | |
| **Lymph-vascular invasion** | | | |
| Negative | 14 (53.8%) | 12 (46.2%) | 0.751 |
| Positive | 7 (43.8%) | 9 (56.2%) | |
| **Perineural invasion** | | | |
| Negative | 19 (48.7%) | 20 (51.3%) | 1.000 |
| Positive | 2 (66.7%) | 1 (33.3%) | |
| **Histology grade** | | | |
| Well or Moderate | 17 (48.6%) | 18 (51.4%) | 1.000 |
| Poorly | 4 (57.1%) | 3 (42.9%) | |
| **Hepatitis B** | | | |
| Negative | 5 (41.7%) | 7 (58.3%) | 0.734 |
| positive | 16 (53.3%) | 14 (46.7%) | |
| **Hepatitis C** | | | |
| Negative | 17 (56.7%) | 13 (43.3%) | 0.306 |
| Positive | 4 (33.3%) | 8 (66.7%) | |
| **Liver cirrhosis** | | | |
| No cirrhosis | 7 (63.6%) | 4 (36.4%) | 0.484 |
| Cirrhosis | 14 (45.2%) | 17 (54.8%) | |
| **CEA** | | | |
| ≤5 | 21 (53.8%) | 18 (46.2%) | 0.232 |
| >5 | 0 (0%) | 3 (100%) | |
| **CA-199** | | | |
| ≤ 35 | 16 (53.3%) | 14 (46.7%) | 0.734 |
| >35 | 5 (41.7%) | 7 (58.3%) | |
| **AFP** | | | |
| ≤ 15 | 9 (47.4%) | 10 (52.6%) | 0.757 |
| >15 | 12 (52.2%) | 11 (47.8%) | |
| **IHC percentage** | | | |
| ≤55% | 14 (60.9%) | 9 (39.1%) | 0.121 |
| >55% | 7 (36.8%) | 12 (63.2%) | |

**Table 3. Correlation between the clinicopathological features and 2-year recurrence-free survival and overall survival in combined hepatocellular cholangiocarcinoma.**

| Variables | No. of patients | Cumulative | P | Cumulative | P |
|---|---|---|---|---|---|
| | | 2- year recurrence-free survival rate | | 5-year overall survival rate | |
| **Age** | | | | | |
| ≦60 | 22 | 46.5% | 0.935 | 37.8% | 0.493 |
| >60 | 20 | 52.5% | | 23.2% | |
| **Sex** | | | | | |
| male | 29 | 44.8% | 0.232 | 29.2% | 0.627 |
| female | 13 | 60.6% | | 57.7% | |
| **AJCC staging** | | | | | |
| I-II | 34 | 58.8% | *<0.001** | 44.2% | *<0.001** |
| III-IV | 8 | 12.5% | | 0% | |
| **T stage** | | | | | |
| T1-2 | 36 | 57.7% | *<0.001** | 41.2% | *<0.001** |
| T3-4 | 6 | 0% | | 0% | |
| **N stage** | | | | | |
| negative | 39 | 51.2% | 0.198 | 37.9% | *<0.001** |
| positive | 3 | 33.3% | | 0% | |
| **Lymph-vascular invasion** | | | | | |
| Negative | 26 | 62.8% | *0.035** | 49.1% | *0.001** |
| Positive | 16 | 29.2% | | 12.9% | |
| **Perineural invasion** | | | | | |
| Negative | 39 | 50.5% | 0.498 | 36.7% | 0.162 |
| Positive | 3 | 33.3% | | 0% | |
| **Histology grade** | | | | | |
| Well or Moderate | 35 | 48.1% | 0.853 | 38.5% | 0.867 |
| Poorly | 7 | 57.1% | | 0% | |
| **Hepatitis B** | | | | | |
| Negative | 12 | 48.6% | 0.882 | 41.9% | 0.947 |
| Positive | 30 | 49.3% | | 34.6% | |
| **Hepatitis C** | | | | | |
| Negative | 30 | 51.7% | 0.865 | 35.3% | 0.805 |
| Positive | 12 | 42.8% | | 34.6% | |
| **Liver cirrhosis** | | | | | |
| No cirrhosis | 11 | 54.5% | 0.833 | 51.9% | 0.465 |
| Cirrhosis | 31 | 47.3% | | 34.2% | |
| **CEA** | | | | | |
| ≤5 | 39 | 51.7% | 0.112 | 36.8% | *0.012** |
| >5 | 3 | 0% | | 0% | |
| **CA-199** | | | | | |
| ≤ 35 | 30 | 50.4% | 0.776 | 45.1% | 0.205 |
| >35 | 12 | 45.7% | | 0% | |
| **AFP** | | | | | |
| ≤ 15 | 19 | 41.5% | 0.788 | 23.3% | 0.303 |
| >15 | 23 | 55.2% | | 41.9% | |
| **NLR** | | | | | |
| <3 | 21 | 84.0% | *0.001** | 60.0% | *0.001** |
| ≧3 | 21 | 9.2% | | 10.8% | |
| **IHC cell percentage** | | | | | |
| ≤55% | 23 | 61.4% | *0.031** | 45.9% | *0.034** |
| >55% | 19 | 35.1% | | 17.2% | |

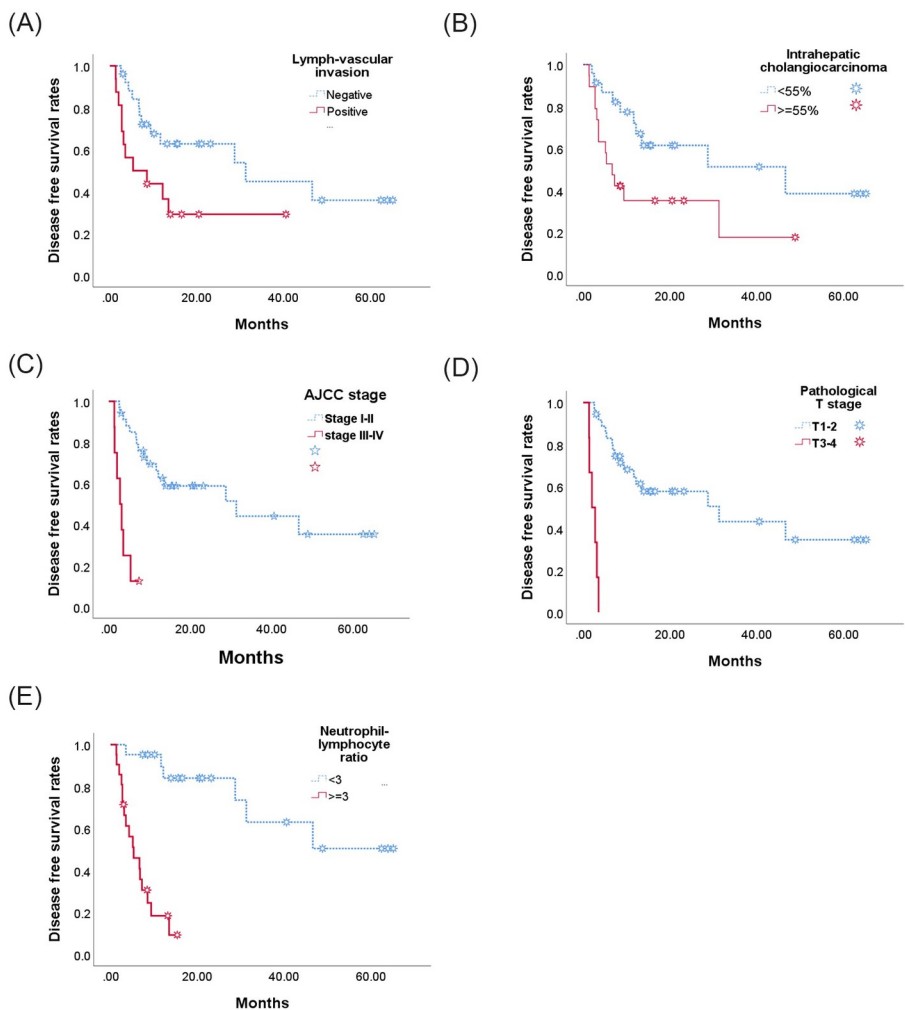

**Fig 1. Kaplan-Meier estimates of the probability of disease free survival (DFS).** Positive Lymph-vascular invasion (LVI)(1a), intratumor cholagnocarcinoma $\geq$ 55% (1b), T-stage III/IV(1c), AJCC tumor stage III/IV(1d) and neutrophil-to-lymphocyte ratio (NLR) $\geq$3 (1e) were associated with poor DFS.

The other interesting finding in our study is that IHC cell predominance (>55%) plays an important prognostic factor of DFS and OS of CHCC-CC patients post tumor resection in Kaplan-Meier univariate analyses. There were just a few studies about predominance of HCC or IHC cells within the tumor in CHCC-CC. Ariizumi et al. reported that CHCC-CC patients had poor outcomes after hepatectomy regardless of the predominance of IHC cells within the tumor. In their study, twenty of forty-four patients (45%) had portal vein invasion and ten patients received transcatheter arterial chemoembolization or radiofrequency ablation before surgery. Portal vein invasion has been found to be a significant poor prognostic factor in HCC [35, 36] and IHC [37] patients receiving operation. Nearly fifty-percent patients with portal vein invasion before surgery may affect their DFS and OS significantly in Ariizumi' study. Although their study showed portal invasion was more frequent in patients with >50% IHC cells within the tumor than in patients with ≤50% IHC cells, but there was no statistically significant difference. Our study just included one patient with portal vein invasion because most were diagnosed as having HCC prior to hepatectomy. At our hospital, HCC with portal vein

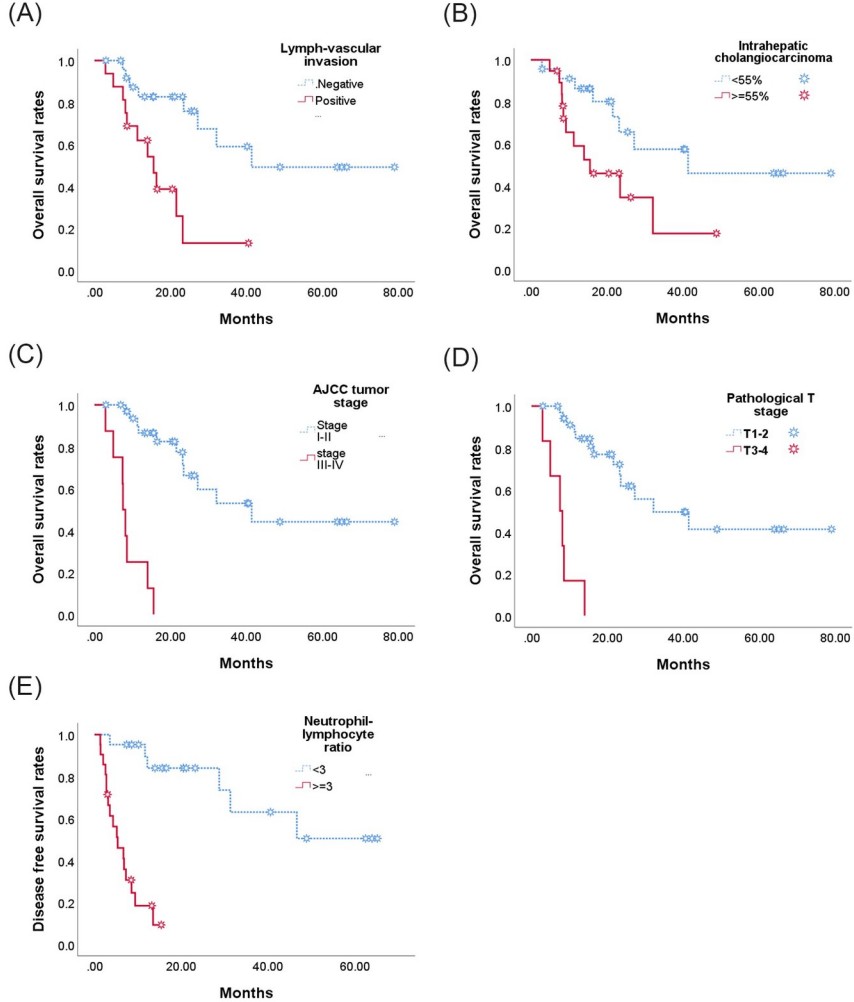

**Fig 2. Kaplan-Meier estimates of the probability of overall survival (OS).** Positive lymph-vascular invasion (LVI) (2a), intratumor cholagnocarcinoma $\geq$ 55% (2b), T stage III/IV (2c), AJCC tumor staging III/VI (2d) and neutrophil-to-lymphocyte ratio (NLR) $\geq$3 (2e) had poorer OS.

**Table 4. Risk factors affecting 2-year recurrence free survival and 5-year overall survival rate determined by Cox regression analysis.**

| Factors | HR | 95%CI | P value |
|---|---|---|---|
| **Disease free survival** | | | |
| Gender female | 0.328 | 0.112–0.967 | 0.043 |
| CA-199 >35 | 7.719 | 1.789–33.316 | 0.006 |
| NLR $\geqq$3 | 41.679 | 7.812–222.378 | 0.001 |
| T stage III-IV | 219.682 | 16.966–2844.518 | 0.001 |
| **Overall survival** | | | |
| NLR $\geqq$3 | 4.148 | 1.196–14.388 | 0.025 |
| AJCC staging III-IV | 10.847 | 3.085–38.140 | 0.001 |

CI, confidence interval; HR, hazard ratios

a. Age, N stage, AJCC tumor stage, lymph-vascular invasion, perineural invasion, histology grade, hepatits B or C, liver cirrhosis, CEA > 5, AFP>15 and intrahepatic cholangiocarcinoma ratio were not associated the recurrence

b. Age, N stage, pathological T stage, lymph-vascular invasion, perineural invasion, histology grade, hepatits B or C, liver cirrhosis, CEA > 5, CA199, AFP>15 and intrahepatic cholangiocarcinoma ratio were not associatedmortality.

**Table 5. Relationships between predominance of intrahepatic cholangiocarcinoma and clinicopathological factors.**

| Intrahepatic cholangiocarcinoma cell | ≤55% | >55% | P |
|---|---|---|---|
| **Age** | | | |
| ≤ 60 | 13 (59.1%) | 9 (40.9%) | 0.551 |
| > 60 | 10 (50%) | 10 (50%) | |
| **Gender** | | | |
| male | 18 (62.1%) | 11 (37.9%) | 0.278 |
| female | 5 (38.5%) | 8 (61.5%) | |
| **AJCC staging** | | | |
| I-II | 21(61.8%) | 13 (38.2%) | 0.112 |
| III-IV | 2 (25%) | 6 (75%) | |
| **T stage** | | | |
| I-II | 22 (61.1%) | 14 (38.9%) | 0.075 |
| III-IV | 1 (16.7%) | 5 (83.3%) | |
| **N stage** | | | |
| Negative | 21 (53.8%) | 18 (46.2%) | 1.000 |
| positive | 2 (66.7%) | 1 (33.3%) | |
| **Lymph-vascular invasion** | | | |
| Negative | 17 (65.4%) | 9 (34.6%) | 0.149 |
| Positive | 6 (37.5%) | 10 (62.5%) | |
| **Perineural invasion** | | | |
| Negative | 22 (56.4%) | 17 (43.6%) | 0.581 |
| Positive | 1 (33.3%) | 2 (66.7%) | |
| **Histology grade** | | | |
| Well or Moderate | 18 (51.4%) | 17 (48.6%) | 0.428 |
| Poorly | 5 (71.4%) | 2 (28.6%) | |
| **Hepatitis B** | | | |
| Negative | 5 (41.7%) | 7 (58.3%) | 0.462 |
| positive | 18 (60%) | 12 (40%) | |
| **Hepatitis C** | | | |
| Negative | 16 (53.3%) | 14 (46.7%) | 1.000 |
| Positive | 7 (58.3%) | 5 (41.7%) | |
| **Liver cirrhosis** | | | |
| No cirrhosis | 4 (36.4%) | 7 (63.6%) | 0.283 |
| Cirrhosis | 19 (61.3%) | 12 (38.7%) | |
| **CEA** | | | |
| ≤5 | 21 (53.8%) | 18 (46.2%) | 1.000 |
| >5 | 2 (66.7%) | 1 (33.3%) | |
| **CA-199** | | | |
| ≤ 35 | 17 (56.7%) | 13 (43.3%) | 0.961 |
| >35 | 6 (50%) | 6 (50%) | |
| **AFP** | | | |
| ≤ 15 | 12 (63.2%) | 7 (36.8%) | 0.320 |
| >15 | 11 (47.8%) | 12 (52.2%) | |
| **NLR** | | | |
| <3 | 14 (66.7%) | 7 (33.3%) | 0.121 |
| ≧3 | 9 (42.9%) | 12 (57.1%) | |

invasion is a contra-indication for surgery. It is difficult to diagnose CHCC-CC accurately before surgery, because CT or MRI scans often do not show typical patterns of contrast uptake or washout. CHCC-CC may have features of both HCC and IHC when a hepatic tumor contains an area of hyper-enhancement in the early phase and an area of delayed enhancement in the late phase on dynamics computed tomography [38].

Survival and prognosis of CHCC-CC patients after hepatectomy varies. Depending on study, 5-year survival rates range from 0% to 62% [39]. The difference in medial survival of those studies may be due to the small number of patients and may not accurately represent the actual prognosis of patients with CHCC-CC. In the current study, median DFS and OS were 16.6 months and 22.6 months, respectively, indicating patients with CHCC-CC had poor OS and DFS despite the curative resection. Most studies have found survival rates to be poorer in CHCC-CC than in HCC and undecided with regard to IHC [9, 40]. We found that two-year disease free survival in patients with less than 55% and more than 55% IHC cells within the tumor to be 61.4% and 35.1%, respectively, and 5-year overall survival rates to be 45.9% and 17.2%, respectively, both significantly different in univariate Kaplan-Meier survival analyses. A predominance of IHC cells within CHCC-CC was an indicator of a more aggressive tumor, though in our multivariate cox regression analysis, did not find the predominance to significantly affect DFS and OS. We found no clinicopathological factors to be associated with IHC predominance in this study (Table 5). Our univariant Kaplan-Meier survival analyses also found AJCC tumor stage, pathological T stage and lymph-vascular invasion to predict DFS and OS, but our multivariant cox regression analyses did not. These pathological factors have been related to poor prognoses in previous studies [41, 42].

This study has some limitations. First, it was a single center retrospective study. Second, the patient sample number was relatively small and selection bias may affect outcome. Diagnosis of CHCC-CC before surgery was difficult. The preoperative diagnoses of patients in this study were either HCC or IHC. Third, our hospital had no standardized treatment of recurrent CHCC-CC. Different treatment modalities might have affected overall survival.

In conclusion, preoperative NLR ≧3 was significantly predicted poor DFS and OS in resectable CHCC-CC. This readily available pre-operative test might potentially be used in conjunction with such post-operative pathological findings as T tumor stage, AJCC tumor stage, lymph-vascular invasion and IHC percentage to improve assessment of tumor biology and treatment decision-making.

## Supporting information

**S1 Fig. ROC curve of Neutrophil/lymphocyte ratio.** Threshold values were determined using the ROC curves, and the value with the highest sensitivity and specificity was calculated.
(TIF)

**S1 Data. Study dataset.** Analytical dataset used in the study.
(XLSX)

## Acknowledgments

The authors thank Kaohsiung Chang Gung Memorial Hospital Cancer Center for patients' clinical data collection.

## Author Contributions

**Conceptualization:** Tai-Jan Chiu, Yen-Yang Chen.

**Data curation:** Tai-Jan Chiu, Yi-Ju Chen, Fang-Ying Kuo, Yen-Yang Chen.

**Formal analysis:** Tai-Jan Chiu, Yi-Ju Chen, Fang-Ying Kuo.

**Funding acquisition:** Tai-Jan Chiu, Yen-Yang Chen.

**Investigation:** Tai-Jan Chiu, Yen-Yang Chen.

**Methodology:** Tai-Jan Chiu, Yi-Ju Chen, Fang-Ying Kuo.

**Project administration:** Yen-Yang Chen.

**Resources:** Tai-Jan Chiu, Yi-Ju Chen.

**Supervision:** Yen-Yang Chen.

**Validation:** Tai-Jan Chiu, Yi-Ju Chen, Fang-Ying Kuo.

**Visualization:** Yen-Yang Chen.

**Writing – original draft:** Tai-Jan Chiu, Yen-Yang Chen.

**Writing – review & editing:** Tai-Jan Chiu, Yen-Yang Chen.

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
