## [Decision Letter · Decision Letter 0]

13 Jul 2020

PONE-D-20-07323

Elevated neutrophil-to-lymphocyte ratio and predominance of intrahepatic cholangiocarcinoma prediction of poor hepatectomy outcomes in  patients with combined hepatocellular–cholangiocarcinoma

PLOS ONE

Dear Dr. Chen,

Thank you for submitting your manuscript to PLOS ONE. After careful consideration, we feel that it has merit but does not fully meet PLOS ONE’s publication criteria as it currently stands.

Therefore, we invite you to submit a revised version of the manuscript that addresses the points raised by all three reviewers. They have been considered minor, but compulsory revisions.

We look forward to receiving your revised manuscript.

Kind regards,

Aldo Scarpa

Academic Editor

PLOS ONE

Journal Requirements:

2. In the ethics statement in the manuscript and in the online submission form, please provide additional information about the patient records used in your retrospective study, specifically whether all data were fully anonymized before you accessed them.

3. Thank you for stating the following in your manuscript:

'Founding

This study was funded by grant obtained by Dr. Yen-Yang Chen from Kaohsiung Chang Gung Memorial Hospital Taiwan (Grant nos. CMRPG8E0881, CMRPG8F1771/1772). This study was also funded by grants obtained by Dr. Tai-Jan Chiu from Kaohsiung Chang Gung Memorial Hospital Taiwan (Grant CMRPG8E0811/0812/0813, CMRPG8C0531 and CMRPG8D0801).

'The author(s) received no specific funding for this work.'

4. Please upload a copy of Figure 4f and 4g, to which you refer in your text. If the figure is no longer to be included as part of the submission please remove all reference to it within the text.

5. Your ethics statement must appear in the Methods section of your manuscript. If your ethics statement is written in any section besides the Methods, please move it to the Methods section and delete it from any other section. Please also ensure that your ethics statement is included in your manuscript, as the ethics section of your online submission will not be published alongside your manuscript.

6. Please include captions for your Supporting Information files at the end of your manuscript, and update any in-text citations to match accordingly. Please see our Supporting Information guidelines for more information: http://journals.plos.org/plosone/s/supporting-information

Reviewers' comments:

Reviewer's Responses to Questions

**Comments to the Author**

1. Is the manuscript technically sound, and do the data support the conclusions?

Reviewer #1: Yes

Reviewer #2: Partly

Reviewer #3: Yes

2. Has the statistical analysis been performed appropriately and rigorously? 

Reviewer #1: Yes

Reviewer #2: No

Reviewer #3: No

3. Have the authors made all data underlying the findings in their manuscript fully available?

Reviewer #1: No

Reviewer #2: No

Reviewer #3: Yes

4. Is the manuscript presented in an intelligible fashion and written in standard English?

Reviewer #1: Yes

Reviewer #2: Yes

Reviewer #3: Yes

5. Review Comments to the Author

Reviewer #1: Thank you for the privilege of reviewing the paper by Chiu, Chen et al examining N-L ratio (NLR) in patients with resection of combined HCC-Cholangioca (intrahepatic) with a predominance of CCA.

Reference 28 signifies that the finding is not terribly novel, but it was a small study (n=59( The study adds incrementally to the body of literature that supports the utility of NLR as a prognostic marker for recurrence post hepatectomy for IHC, HCC here examining combined HCC-IHC. This is still worthwhile in my opinion since there isn’t a lot of data or large numbers in published studies to date. The angle of IHC predominance is of some interest, but there may be confounding, see below.

Some clarifications would be useful

Did any cases have branch portal vein invasion? If so it has to be included in the descriptive analysis and model

Is there an approved definition of HCC-CCA or predominance of IHC vs HCC in mixed lesions? Consider the paper by Brunt et al PMID: 29360137. I believe the authors have been careful in their case selection but important to add some references here where possible

Where did the cut-offs for Age CEA CA199 AFP NLR and IHC% come from? This would be important to describe, especially for NLR, some descriptive analysis on this in particular is warranted

Did NLR correlate with CA19-9 or IHC percentage? Nice to describe and does not negative the validity of data if there was a relationship, but we have to overtly recognize the risk of confounding then

In Table 4, please provide by footnote the factors not associated with disease recurrence or mortality

For sake of sensitivity analysis how would dose NLR as a continuous or categorical variable predict endpoints when age AFP CA199 CEA and IHC% are included as continuous variables

Figures, formatting needed, change days to years, text too small on axes, image quality poor and add nothing to data shown in tables

Consider a graph for NLR above and below threshold that is adjusted for all other factors, this can be done in Cox-regression in SPSS, that includes all factors, making NLR a categorical variable and clicking plot, follow the prompts to show smoothed survival curves by selected cat variable (NLR). This way you have two graphs with risk adjusted outcomes for recurrence and OS

Reviewer #2: THis is a descriptive, paper describing NLR in intrahepatic cholangiocarcinoma looking at a series of cases collected over 13 years at a single institution

Suggestions: it is stated that cutoff of 3 determined from ROC curve, but curve not shown

The authors suggest that this is a predictive model but is actually a descriptive case series.

Reviewer #3: The manuscript titled “Elevated neutrophil-to-lymphocyte ratio and predominance of intrahepatic cholangiocarcinoma prediction of poor hepatectomy outcomes in patients with combined hepatocellular–cholangiocarcinoma” is a study about prognostic value of elevated neutrophil-to-lymphocyte ratio in surgical treated combined hepatocellular – cholangiocarcinoma. The Authors should be congratulated for collecting a clinical series of 42 patients resected with this rare disease

The manuscripts have the following limitations:

• The Authors should discuss the criteria for dividing patients according to the proportion of IHC cell (more or less than 55%);

• The Authors should clarify the reason for reporting univariate analysis at 2-year for recurrence-free survival and at 5 years for overall survival;

• The Authors should report the rate of positive and negative margins among patients submitted to resection;

• The small sample size and the distribution of different characteristics may have influenced the power of statistical analysis, the authors should comment the HR rate for T stage for recurrent disease (HR= 219.682). I suggest to review the statistical analysis by a statistician.

6. PLOS authors have the option to publish the peer review history of their article (what does this mean?). If published, this will include your full peer review and any attached files.

Reviewer #1: No

Reviewer #2: No

Reviewer #3: **Yes: **Andrea Ruzzenente

---

## [Author Response · Author response to Decision Letter 0]

16 Sep 2020

Response to Editor comments:

1. Please ensure that your manuscript meets PLOS ONE's style requirements, including those for file naming. The PLOS ONE style templates can be found at https://journals.plos.org/plosone/s/file?id=wjVg/PLOSOne_formatting_sample_main_body.pdf and https://journals.plos.org/plosone/s/file?id=ba62/PLOSOne_formatting_sample_title_authors_affiliations.pdf.

Response: We agree with this valuable comment. We will make sure our manuscript to meet PLOS ONE style

2. In the ethics statement in the manuscript and in the online submission form, please provide additional information about the patient records used in your retrospective study, specifically whether all data were fully anonymized before you accessed them.

Response: We agree with this valuable comment. In the “Ethics approval and consent to participate”, we provide Availability of data and materials

The datasets used and/or analyzed during the current study are available from the corresponding author on reasonable request

3. 'Founding

This study was funded by grant obtained by Dr. Yen-Yang Chen from Kaohsiung Chang Gung Memorial Hospital Taiwan (Grant nos. CMRPG8E0881, CMRPG8F1771/1772). This study was also funded by grants obtained by Dr. Tai-Jan Chiu from Kaohsiung Chang Gung Memorial Hospital Taiwan (Grant CMRPG8E0811/0812/0813, CMRPG8C0531 and CMRPG8D0801).

'The author(s) received no specific funding for this work.'

Response: we will remove funding information from text and update our Funding statement.

Response: Our amended statements will be within our cover letter

4. Please upload a copy of Figure 4f and 4g, to which you refer in your text. If the figure is no longer to be included as part of the submission please remove all reference to it within the text.

Response: We correct our mistakes of Figure 4f and 4g to Figure 2f and 2g 

5. Your ethics statement must appear in the Methods section of your manuscript. If your ethics statement is written in any section besides the Methods, please move it to the Methods section and delete it from any other section. Please also ensure that your ethics statement is included in your manuscript, as the ethics section of your online submission will not be published alongside your manuscript.

Response: We rewrite our ethics statement in the Methods section of our manuscript 

Response to Reviewer 1’s Comments:

1. Did any cases have branch portal vein invasion? If so it has to be included in the descriptive analysis and model.

Response: We agreed with these comments. In this study, we just found only one patient had tumor with portal vein invasion (T4). In the section of result, we showed that only one patient had tumor with portal vein invasion. At our hospital, HCC with portal vein invasion is a contra-indication for surgery. It is difficult to diagnose CHCC-CC accurately before surgery, because CT or MRI scans often do not show typical patterns of contrast uptake or washout. The patient number was too few, and we did not include this item in our analysis. However, in our study, there were sixteen patients with microscopic lymph-vascular invasion. We found cHCC-CC patients with lymph-vascular invasion predicted poor outcome after resection. 

2. Is there an approved definition of HCC-CCA or predominance of IHC vs HCC in mixed lesions? Consider the paper by Brunt et al PMID: 29360137. I believe the authors have been careful in their case selection but important to add some references here where possible

 Response: I think this is a very important comment. We citated this paper in our reference [24]. In this retrospective study, we collected medical records of 42 patients with primary CHCC-CC treated with surgical resection between January 2000 and December 2013 at our hospital. Two pathologists performed immunohistochemical studies of tissue samples collected during surgery. Following Allen and Lisa classification, we included only patients with CHCC-CC classified as type C (intimate intermingling of hepatocellular and glandular elements) and type B (contiguous but independent masses of HCC and CC). We excluded patients with type A (separate masses constituting either HCC or CC). CHCC-CC was diagnosed pathologically based on microscope studies of thin-section specimens stained with hematoxylin and eosin. The immunoreactivity of each tumor was confirmed: hepatocyte paraffin 1 (Hep1) antibody and CK-7 (cytokeratin-7) in CHCC-CC. 

3. Where did the cut-offs for Age CEA CA199 AFP NLR and IHC% come from? This would be important to describe, especially for NLR, some descriptive analysis on this in particular is warranted.

Response: We agreed with these comments. 

a. The cut-offs values of CEA, CA199 and AFP were the upper limit of normal range in our hospital. 

b. The cut-off value for Age is the median age of these 42 patients. 

c. Definition of NLR: NLR was calculated by dividing the absolute neutrophil count by the absolute lymphocyte count. NLR was determined within seven days before hepatectomy. Based on the time-dependent receiver operating characteristic (ROC) curve of our study, we set the NLR cutoff value to be 3.0(NLR < 3, NLR ≥ 3). Previous studies in intrahepatic cholangiocarcinoma have used a similar threshold[25, 26].

d. IHC cell predominance (>55%): The definition was discussed with pathologists with expertise in hepatic tumors. However, there was no related reference. Ariizumi et al. reported that CHCC-CC patients had poor outcomes after hepatectomy regardless of the predominance of IHC cells within the tumor. Their definition of IHC predominance was >50% IHC cells within the tumor. 

4. Did NLR correlate with CA19-9 or IHC percentage? Nice to describe and does not negative the validity of data if there was a relationship, but we have to overtly recognize the risk of confounding then

 Response: We agreed with these comments. Table 2 showed relationships between elevated neutrophil-to-lymphocyte ratio and clinicopathological factors. We did not find the relationship between NLR and CA199 or IHC predominance. 

5. In Table 4, please provide by footnote the factors not associated with disease recurrence or mortality.

 Response: We agreed with these comments. We provided by footnote the factors not associated with disease recurrence or mortality. 

6. For sake of sensitivity analysis how would dose NLR as a continuous or categorical variable predict endpoints when age AFP CA199 CEA and IHC% are included as continuous variables.

 Response: We agreed with these comments. In Table 2, we found that there was no relationship between NLR and age, AFP, CA199, CEA and IHC%. NLR could predict tumor recurrence and survival time although cox-regression analyses including age, AFP, CA199, CAE, AFP and IHC%

Recurrence 

 B 標準誤差 Wald 自由度 顯著性 Exp(B) 95.0% Exp(B) 的 CI

 下限 上限

Age60 -.227 .446 .258 1 .611 .797 .333 1.911

NLR2 2.893 .736 15.431 1 .000 18.039 4.260 76.383

CEA1 .575 .820 .492 1 .483 1.778 .356 8.872

CA199a -.559 .567 .970 1 .325 .572 .188 1.738

AFP1 -.187 .467 .160 1 .689 .829 .332 2.073

Cho55 .901 .502 3.217 1 .073 2.461 .920 6.587

Overall survival 

 B 標準誤差 Wald 自由度 顯著性 Exp(B) 95.0% Exp(B) 的 CI

 下限 上限

Age60 -.754 .505 2.225 1 .136 .471 .175 1.267

NLR2 1.972 .652 9.149 1 .002 7.187 2.002 25.795

CEA1 1.440 .851 2.865 1 .090 4.223 .797 22.383

CA199a .738 .519 2.019 1 .155 2.091 .756 5.784

AFP1 -1.013 .531 3.647 1 .056 .363 .128 1.027

Cho55 1.338 .555 5.819 1 .016 3.812 1.285 11.305

7. Figures, formatting needed, change days to years, text too small on axes, image quality poor and add nothing to data shown in tables 

Response: We agree with this valuable comment and we re-format the figures

2). How many pathologist was read IHC data? What criteria to accept on grading score from different reader?

Response: In the “immunohistochemical study”, MK immunostaining was evaluated independently by two pathologists blinded to the subjects’ clinical information. Each specimen was assigned a score of 1 to 4 based on the percentage of positive cells within a field of cells (100 x magnification): 1 for <5% of the cells, 2 for 6–35% of the cells, 3 for 36–70% of the cells, and 4 for >71% of the cells. Each specimen also received another score of 1 to 4 based on intensity of staining: 1 for negative staining, 2 for weak staining, 3 for moderate staining and 4 for strong staining. MK expression score was then calculated by multiplying the percentile and intensity scores. A score of ≥4 for MK protein expression levels indicated the tumor was positive

Response to Reviewer 2’s Comments:

it is stated that cutoff of 3 determined from ROC curve, but curve not shown Response: We agree with this valuable comment and we provide our ROC curve at supplementary Figure 1

Response to Reviewer 3’s Comments:

1. The Authors should discuss the criteria for dividing patients according to the proportion of IHC cell (more or less than 55%)

Response: We agreed with these comments. However, from the literature review, there was no study to evaluate whether the percentage of intra-hepatic cholaginocarcinoma affect the outcome in CHCC-CC patients. After discussion with our pathologist with expertise in hepatic tumors, we made the definition of predominance of intrahepatic cholangiocarcinoma while those whose IHC cells made up more than 55%

2. The Authors should clarify the reason for reporting univariate analysis at 2-year for recurrence-free survival and at 5 years for overall survival;

Response: We agreed with these comments. Survival and prognosis of CHCC-CC patients after hepatectomy varies. Depending on study, 5-year survival rates range from 0% to 62%[39]. In the current study, Mean survival follow-up was 677 days (87-2371 days). Nineteen of the patients died during the course of this study. Mean disease free survival was 499 days, two-year recurrence free survival was 43.2%, and five-year overall survival 34.9%. Therefore, we report the univariate analysis at 2- year for recurrence free survival and 5 years for overall survival. 

3. The Authors should report the rate of positive and negative margins among patients submitted to resection.

Response: We agree with this valuable comment. We reported our surgical margin (< 1cm or � 1cm) on Table 1. Eighteen patients (42.8%) had surgical margin < 1 cm. 

4. The small sample size and the distribution of different characteristics may have influenced the power of statistical analysis, the authors should comment the HR rate for T stage for recurrent disease (HR= 219.682). I suggest to review the statistical analysis by a statistician. 

Response: We agree with this valuable comment. We discussed with the statistician. They thought that HR rate for pathological T stage for recurrence was affected significantly because the patient number of pathological T3-4 was only 6 patients. The sample size was too small and all these patients got recurrence within 4 months (40 days to 106 days).

---

## [Decision Letter · Decision Letter 1]

5 Oct 2020

Elevated neutrophil-to-lymphocyte ratio and predominance of intrahepatic cholangiocarcinoma prediction of poor hepatectomy outcomes in  patients with combined hepatocellular–cholangiocarcinoma

PONE-D-20-07323R1

Dear Dr. Chen,

We’re pleased to inform you that your manuscript has been judged scientifically suitable for publication and will be formally accepted for publication once it meets all outstanding technical requirements.

Kind regards,

Aldo Scarpa

Academic Editor

PLOS ONE

Reviewers' comments:

Reviewer's Responses to Questions

**Comments to the Author**

1. If the authors have adequately addressed your comments raised in a previous round of review and you feel that this manuscript is now acceptable for publication, you may indicate that here to bypass the “Comments to the Author” section, enter your conflict of interest statement in the “Confidential to Editor” section, and submit your "Accept" recommendation.

Reviewer #1: All comments have been addressed

Reviewer #3: All comments have been addressed

2. Is the manuscript technically sound, and do the data support the conclusions?

Reviewer #1: Yes

Reviewer #3: Yes

3. Has the statistical analysis been performed appropriately and rigorously? 

Reviewer #1: Yes

Reviewer #3: Yes

4. Have the authors made all data underlying the findings in their manuscript fully available?

Reviewer #1: Yes

Reviewer #3: Yes

5. Is the manuscript presented in an intelligible fashion and written in standard English?

Reviewer #1: Yes

Reviewer #3: Yes

6. Review Comments to the Author

Reviewer #1: The authors have addressed my comments, other reviewer comments seem less comprehensively approached and overall there are minimal changes to the paper, nevertheless the data is what it is and for a purely descriptive study it is reasonable to publish

Reviewer #3: The Authors addressed the comments of the reviewers, and they imporved the manuscript accordignto the suggestions of the reviewers

7. PLOS authors have the option to publish the peer review history of their article (what does this mean?). If published, this will include your full peer review and any attached files.

Reviewer #1: No

Reviewer #3: **Yes: **Andrea Ruzzenente

---

## [Editor Report · Acceptance letter]

2 Dec 2020

PONE-D-20-07323R1 

Elevated neutrophil-to-lymphocyte ratio and predominance of intrahepatic cholangiocarcinoma prediction of poor hepatectomy outcomes in patients with combined hepatocellular–cholangiocarcinoma 

Dear Dr. Chen:

I'm pleased to inform you that your manuscript has been deemed suitable for publication in PLOS ONE. Congratulations! Your manuscript is now with our production department. 

Kind regards, 

on behalf of

Dr. Aldo Scarpa 

Academic Editor

PLOS ONE